# Experimental Test and Prediction Model of Soil Thermal Conductivity in Permafrost Regions

**Fu-Qing Cui [1] , Zhi-Yun Liu [1],\*, Jian-Bing Chen [2], Yuan-Hong Dong [2], Long Jin [2] and Hui Peng [2]**

1. College of Geology Engineering and Geomatics, Chang'an University, Xi'an 710054, China; cfq731@chd.edu.cn
2. State Key Laboratory of Road Engineering Safety and Health in Cold and High-Altitude Regions, CCCC First Highway Consultants Co. Ltd., Xi'an 710065, China; chenjb@ccroad.com.cn (J.-B.C.); dyhvic@gmail.com (Y.-H.D.); jlcoolmai@163.com (L.J.); penghui1230@163.com (H.P.)
* Correspondence: Dcdgx33@chd.edu.cn; Tel.: +86-13484463421

**Abstract:** Soil thermal conductivity is a dominant parameter of an unsteady heat-transfer process, which further influences the stability and sustainability of engineering applications in permafrost regions. In this work, a laboratory test for massive specimens is performed to reveal the distribution characteristics and the parameter-influencing mechanisms of soil thermal conductivity along the Qinghai–Tibet Engineering Corridor (QTEC). Based on the measurement data of 638 unfrozen and 860 frozen soil specimens, binary fitting, radial basis function (RBF) neural network and ternary fitting (for frozen soils) prediction models of soil thermal conductivity have been developed and compared. The results demonstrate that, (1) particle size and intrinsic heat-conducting capacity of the soil skeleton have a significant influence on the soil thermal conductivity, and the typical specimens in the QTEC can be classified as three clusters according to their thermal conductivity probability distribution and water-holding capacity; (2) dry density as well as water content sometimes does not have a strong positive correlation with thermal conductivity of natural soil samples, especially for multiple soil types and complex compositions; (3) both the RBF neural network method and ternary fitting method have favorable prediction accuracy and a wide application range. The maximum determination coefficient ($R^2$) and quantitative proportion of relative error within ±10% ($P_{\pm 10\%}$) of each prediction model reaches up to 0.82, 0.88, 81.4% and 74.5%, respectively. Furthermore, because the ternary fitting method can only be used for frozen soils, the RBF neural network method is considered the optimal approach among all three prediction methods. This study can contribute to the construction and maintenance of engineering applications in permafrost regions.

**Keywords:** soil thermal conductivity; water holding capacity; RBF neural network; ternary fitting method; Qinghai-Tibet Engineering Corridor

## 1. Introduction

A rammed soil foundation is a widely utilized construction and building foundation type in permafrost regions and its thermal response to external heat disturbance determines the stability and sustainability of engineering applications to a certain extent [1,2]. Due to the temperature sensitivity of frozen soil, the mechanical properties differ greatly before and after freezing and thawing, which tends to induce various engineering diseases characterized by thaw settlement and frost heave [3]. Thermal conductivity is one of the most important parameters of frozen soil, which is also a prime influencing factor of engineering structural damage caused by thaw settlement and frost heave [4,5]. As a complex multi-phase composition, previous research indicates that the thermal conductivity of frozen soil is associated with many factors [6–8]: inherent mineralogical skeleton, particle size, spatial

pore arrangement, porosity, dry density, water content and temperature, etc. Thus, investigation of the variation features and parameter influence mechanism of soil thermal conductivity is of great importance and contributes to the promotion of stability and sustainability of engineering applications in cold regions.

In past decades, a number of experimental measurements and calculation models were utilized to obtain the thermal conductivity of soil [4–10]. The experimental measurement method can be roughly divided into two categories: steady state method and transient method [4,6,9,10]. The corresponding methods includes steady-state heat flow meter, steady-state comparison, steady-state flat plate, transient spherical probe, transient hot wire, transient heat pulse and transient plane source (TPS) methods etc. [11–14]. For instance, Xu et al. [15] measured the thermal conductivity of the frozen and unfrozen soils using the steady state comparative method, and proposed some widely used empirical formulas between thermal conductivity and influencing factors' parameters (e.g., dry density and water content). Alrtimi et al. [16] designed an improved steady state apparatus for measuring soil thermal conductivity, and a thermal jacket was utilized in the experiment to minimize the radial heat loss. Lu et al. [17], Li et al. [18], and Zhang et al. [19] conducted a series of laboratory test using the transient hot wire method to investigate aeolian sand, frozen clay, and silty clay, respectively. Kojima et al. [20] measured the thermal properties of partially frozen soil with a dual probe heat-pulse sensor, and a numerical solution for radial heat conduction was developed to determine the thermal parameters accurately and simultaneously.

Furthermore, due to high cost, time-consuming and usually limited conditions of experimental measurement, many researchers also summarized a considerable number of calculation models of soil thermal conductivity with different methods [7,8]. Mickley [21] presented a semi-empirical predictive model for moist soils and testified that soil thermal conductivity could be estimated reasonably using its dry density and water content. Cheng et al. [22] developed a theoretical model to predict the thermal conductivity of heterogeneous solid mixtures assuming the parabolic distribution of discontinuous phase volume fraction. Johansen [23] proposed a normalized thermal conductivity conception (Ke, which was the function of thermal conductivity of soils, dry soils and saturated soils) and given the relationship between Ke and saturation level (Sr) for soil thermal conductivity interpolated calculation. Recently, based on a large number of experimental results, Gangadhara et al. [24] established an empirical formula of relative porosity, saturation degree, dry density and water content to predict thermal conductivity. Côté et al. [25] and Lu et al. [26] further studied the thermal conductivity of soils and established a new Ke–Sr relationship incorporating variables of soil type and porosity separately. Nicolsky et al. [27] and Jafarov et al. [28] proposed an inverse modeling method to recover soil thermal conductivity using time series of measured data and physics-based numerical models. Likewise, other researchers also proposed some improved empirical models for wider application by introducing more soil parameters, such as soil texture-dependent parameters, quartz content, fluid network connectivity, etc. [6,29–33]. Considering the effects of freeze-thaw process on the thermal conduction behavior, many scholars discussed the influence of multiple freeze-thaw cycles on the thermal conductivity of frozen soil and developed some useful models to estimate thermal conductivity under temperature and phase transition condition [34–37]. Meanwhile, the statistical-physical method [4,38], artificial neural network method [39] and mesoscopic predictions method [40,41] were also applied for the prediction of soil thermal conductivity.

The Qinghai–Tibet Engineering Corridor (QTEC) is defined as the engineering channel from Golmud to Lhasa in the western China, which includes five major linear projects: Qinghai–Tibet Highway, Qinghai–Tibet Railway, Golmud–Lhasa oil pipeline, Lanzhou–Xining–Lhasa optical cable communication project and high-voltage power transmission and transformation project [42]. It is the widest continues permafrost distribution region in the high altitudes around the world, and with changeable terrain structure, frequent geological disasters, cold and hypoxic climate, its engineering geological conditions are extremely harsh [43,44]. With global warming, the permafrost in the corridor is in a dynamic equilibrium or even gradual degradation [1–3,45]. However, due to the complex

geological distribution and sampling restrictions, the research on the thermal conductivity of various soil types of permafrost regions in the QTEC is not systematic and comprehensive, which needs to be further investigated. Therefore, in the present work, the variation features of the thermal conductivity of massive drilling specimens along the QTEC have been analyzed. Then, based on the cluster analysis of typical soils, the distribution characteristics of dry density, water content and thermal conductivity of three soil clusters have been investigated, and three prediction models have been developed and compared to obtain the optimal calculation models of soil thermal conductivity.

## 2. Research Significance

Permafrost is widely distributed along the QTEC, which is sensitive to external thermal disturbance [1]. With the engineering structures constructed, the pre-existing hydrothermal balance of natural foundation has been seriously disturbed. Coupled with frequent freeze-thaw cycles, the thermal imbalance of engineering structures will lead to uneven degradation of the underlying permafrost, then resulting in different degrees of engineering diseases [2–5]. Thermal conductivity is a prime influence factor of permafrost response to the external thermal disturbance and also a determining parameter of the freeze-thaw circle, which governs the unsteady heat-transfer process of soil [7,8]. Therefore, the comprehensive acknowledgement on the soil thermal conductivity variation law along the QTEC is a prerequisite of design and construction of the engineering structures in this area.

As far as the authors know, this paper proposes for the first time to use massive specimens sourced from QTEC to reveal a variation law of soil thermal conductivity and their parameter-influencing mechanisms, and also develops reasonable prediction models for different temperature conditions. The research results of this paper can contribute to the construction and maintenance of engineering applications in cold regions.

## 3. Testing Scheme

### 3.1. Test Sample

#### 3.1.1. Source of Test Samples

The testing specimens are sourced from the Qinghai–Tibet expressway geological exploration project, and obtained by the drilling sampling method. As shown in Figure 1, the sampling spots are from the Xidatan to Tanggula mountain (the corresponding mileage of present Qinghai-Tibet Highway is K2870~K3307), which is mainly in the permafrost regions of QTEC. The drilling operation is implemented by truck-mounted geological exploration rig, and the diameter of the sampler is 108 mm. After being taken out from the sampler, the soil sample is immediately loaded into a 0.6 m height-marked sample vessel to avoid external disturbance.

The total testing number of soil samples was 1060, which included 638 unfrozen specimens and 860 frozen specimens. The maximum sampling depth was 40 m, and each drilling hole was sampled for different depths. The detail sampling depth distribution is shown in Table 1.

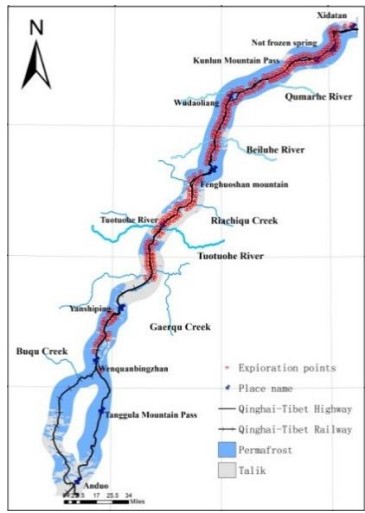

**Figure 1.** Schematic of the sampling spot along the Qinghai–Tibet Engineering Corridor (QTEC).

**Table 1.** Sampling depth distribution of testing specimens in the QTEC.

| Sampling Depth/m | Samples Number | Soil Types |
|---|---|---|
| ≤1 | 73 | clay, silty clay, silt, silty sand, weathered slate, medium fine sand, gravel sand, boulder, breccia, gravel |
| 1~2 | 166 | clay, silty clay, silt, silty sand, weathered mudstone, weathered phyllite, weathered slate, medium fine sand, gravel sand, boulder, breccia, gravel |
| 2~3 | 142 | Same as above |
| 3~5 | 197 | Same as above |
| 5~10 | 183 | Same as above |
| 10~20 | 220 | clay, silty clay, silt, silty sand, weathered mudstone, weathered phyllite, weathered slate, medium fine sand, gravel sand, boulder, breccia, gravel, sandy pebble |
| >20 | 79 | clay, silty clay, silt, silty sand, weathered mudstone, weathered phyllite, weathered slate, medium fine sand, gravel sand, boulder, breccia |

### 3.1.2. Soil Types of Testing Samples

To guarantee the experiment's progress and data representativeness, the specimens with large dry density, high water content, and number of samples less than 15 were not measured in the test. According to the classification criteria of geotechnical engineering, the remaining 907 specimens were identified as 12 types (see Table S1 in Supplementary Materials). The statistic numbers of test samples for certain soil types are shown in Figure 2. It can be seen from the figure that silty clay is the dominant soil type in the QTEC (>30%), and the following is sandy and gravel soil (about 20%), which is basically equivalent.

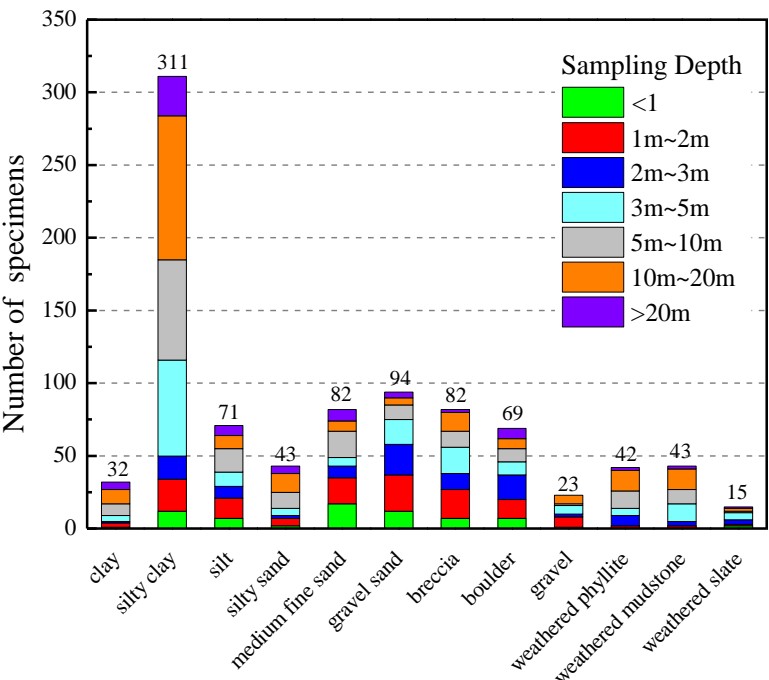

**Figure 2.** Statistic numbers of test samples for certain soil type.

### 3.2. Experimental Methods

A Hot Disk 1500s Thermal Conductivity Analyzer (Hot Disk Co. Ltd., Uppsala, Sweden) was used to measure the thermal conductivities of all specimens (Figure 3a). The analyzer's measurement accuracy is ±3%, which is based on the transient plane source (TPS) testing method. The isotropic hypothesis was utilized in the thermal conductivity measurement. During the test, the analyzer obtained temperature variation values ($\Delta\overline{T}(\tau)$) and the dimensionless time constant ($D(\tau)$) by monitoring the change of film sensor resistance, then drew the corresponding relationship curve, and finally calculated soil thermal conductivity ($\lambda$) by the slope value of $\Delta\overline{T}(\tau)$ and $D(\tau)$ line equations:

$$\Delta\overline{T}(\tau) = \frac{P_0}{\pi^{3/2}r\lambda}D(\tau)$$ 　　　(1)

where $P_0$ is the total output of power and $r$ is the radius of the film sensor.

In order to make the specimen as large as possible, the No. 4922 Kapton film probe was used in the experiment (radius is 29.4 mm). The specimens were formed into two columns with 80 mm in diameter and 30 mm in height. The dry density and water content of specimens were determined by the field measurement data, which was tested by the suspending weigh method and oven drying method, respectively. The corresponding test soil was taken from sample library (Figure 3b). After drying, milling and sieving, the specimen was sampled by the statistic pressure method. Furthermore, for the high water content and icy specimens, the crushed smoothie was used for specimen making.

In the test, the Kapton probe was fixed on the stainless steel sample holder, and two sample columns were placed on both sides of the probe (Figure 3c). To ensure the probe was in close contact with the sample, two plastic film-covered specimens were compacted with a sample holder's clamp. As the measurement of unfrozen soil thermal conductivity finished, the specimens were placed in a freezing cabinet (Figure 3d) and frozen for 24 h ($T < -10$ °C). Then the specimens were placed into the stainless steel sample holder and continued to freeze in the cabinet for 2 h again (to avoid the environment thermal disturbance). The thermal conductivity analyzer needs to define appropriate test time and power parameters to obtain reasonable test results. For every specimen, the thermal conductivity measurement was conducted twice and took the average of the two test results as the experimental result. When the measuring error was larger than 5% due to improper parameter settings,

multiple experiments were conducted to obtain reasonable test results. The main test procedures are listed in Figure 4. In addition, the flatness of the pre-frozen soil sample measuring surface was necessarily checked before the testing of frozen soil. The roughness surface needed to be polished by scraper and placed in the cabinet until the temperature fluctuation was eliminated.

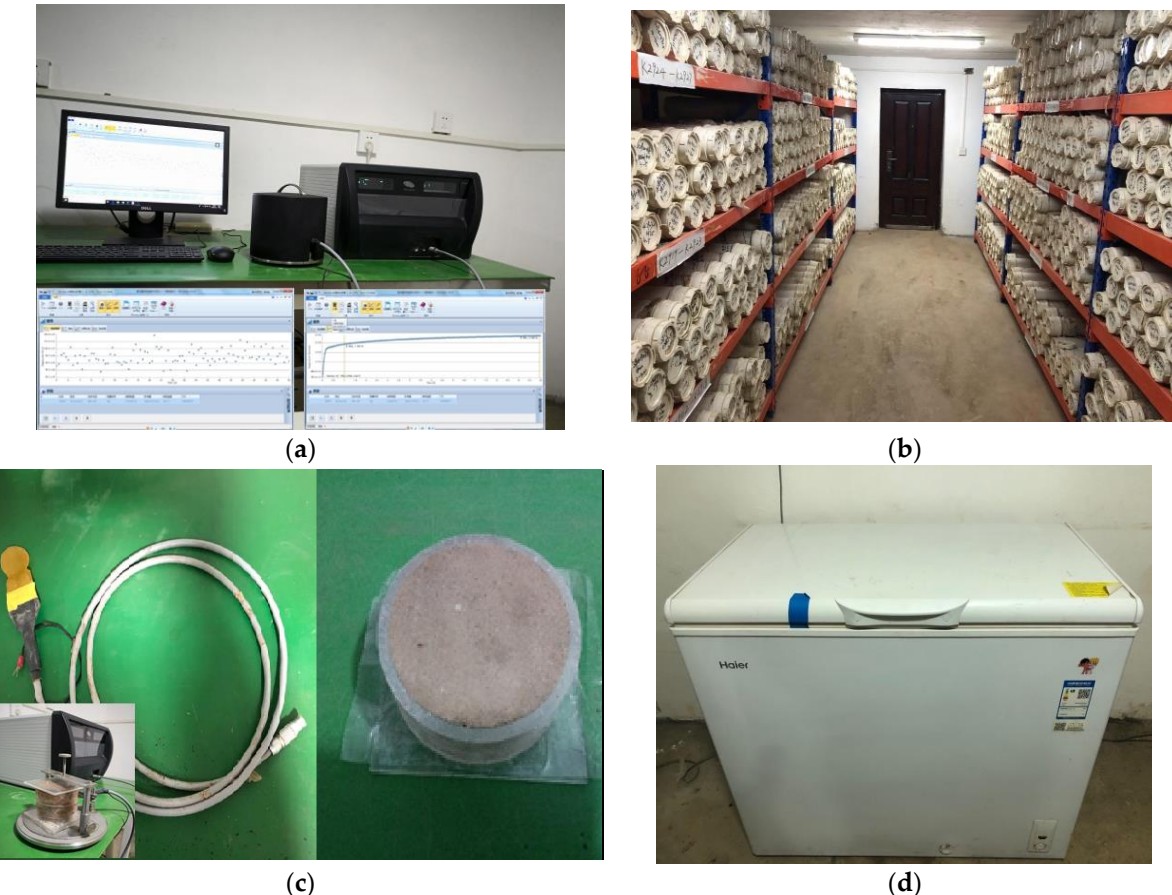

**Figure 3.** Schematic of thermal conductivity test system. (**a**) Thermal conductivity analyzer; (**b**) soil sample library; (**c**) specimen and Kapton probe; (**d**) freezing cabinet.

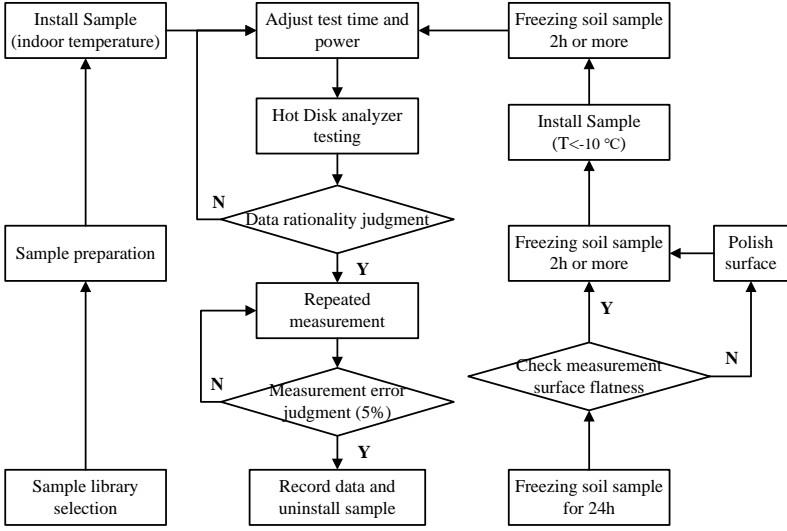

**Figure 4.** Flow diagram of thermal conductivity test.

## 4. Distribution Characteristics and Cluster Analysis of Frozen and Unfrozen Soil Thermal Conductivities

### 4.1. Cluster Analysis of Different Soils' Thermal Conductivity

The probability distribution of thermal conductivity of 12 types of unfrozen soil ($\lambda_u$) is shown in Figure 5. It can be seen in Figure 5a that the thermal conductivity of clay and silty clay soil is significantly smaller than that of other soil types in the high probability interval. Comparatively, the thermal conductivity of clay and silty clay soil in their main distribution range (probability greater than 10%) is between 1.1~1.7 W/(m · K), while the thermal conductivity of weathered mudstone varies in the range of 1.3~1.9 W/(m · K). The average thermal conductivity values of two clay soils and weathered mudstone are 1.415 and 1.513 W/(m · K), respectively. It also shows in Figure 5b that the main probability distributions of thermal conductivity of medium fine sand, gravel sand, boulder, breccia and gravel are all at the intervals of 1.5~2.3 W/(m · K), which is larger than that of the silty sand. Moreover, according to the comparative results of Figure 5a,b, it can be concluded that silt, silty sand, weathered mudstone, weathered phyllite and weathered slate have similar thermal conductivity probability distribution patterns. This indicates that particle size and intrinsic heat-conducting capacity of the soil skeleton have a significant effect on soil thermal conductivity. As the particle size of soil tends to be finer, the average heat transfer cross-section area reduces and the thermal resistance increases, which results in decreasing soil thermal conductivity. Meanwhile, as the thermal conductivity of soil's mineral skeleton increases, the heat-transfer efficiency along the heat-transfer path improves either, which results in the increasing of soil thermal conductivity. Therefore, compared to gravel soil or sandy soil, the thermal conductivity of clay soil is usually relatively small due to its fine particle size and lower soil skeleton thermal conductivity.

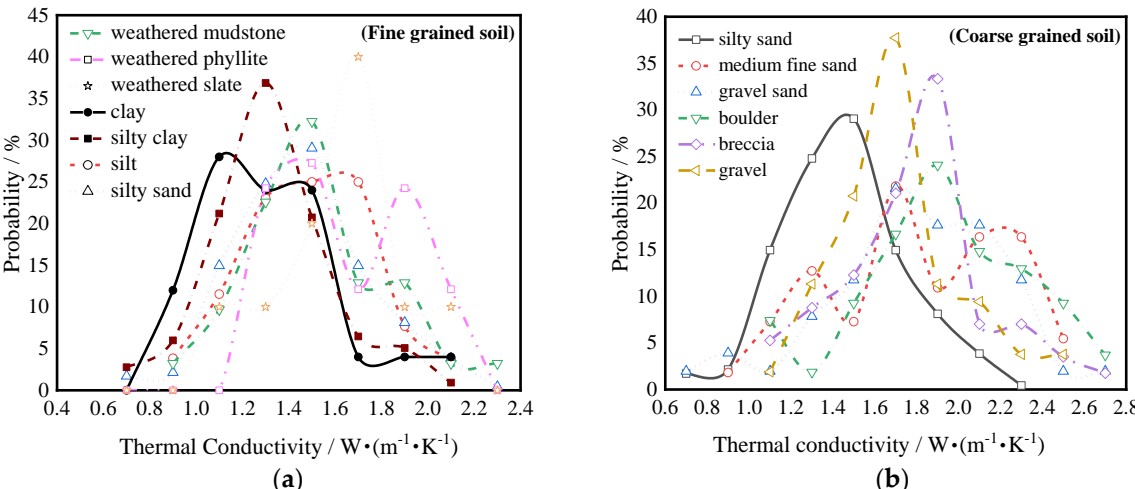

**Figure 5.** Probability distribution of thermal conductivity of 12 types of unfrozen soil. (**a**) Fine grained soil, (**b**) Coarse grained soil.

Figure 6 shows the thermal conductivity probability distribution of 12 types of frozen soil ($\lambda_f$). It can be seen that the main thermal conductivity distribution intervals of frozen silt, silty sand, weathered mudstone, weathered phyllite and weathered slate are basically between 1.7 and 2.5 W/(m · K) (Figure 6a), which confirms the reasonableness of the above classification. Likewise, the thermal conductivity distribution law of frozen clay soil (clay and silty clay) or sand-gravel soil (medium fine sand, gravel sand, boulder, breccia and gravel in Figure 6b) can also be classified into one cluster for analysis.

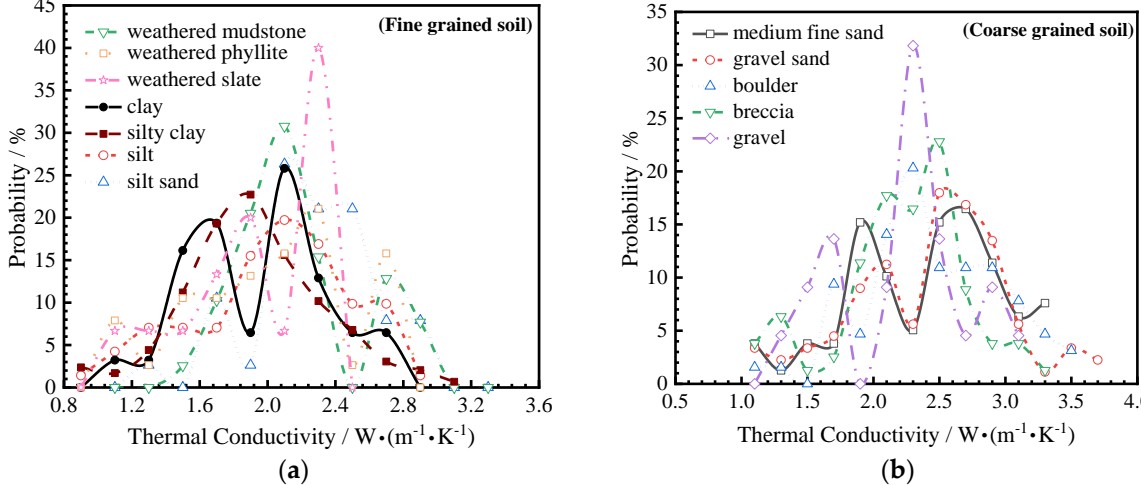

**Figure 6.** Probability distribution of thermal conductivity of 12 types of frozen soil. (**a**) Fine grained soil, (**b**) Coarse grained soil.

Furthermore, considering the difference of soil water holding capacity, the above 12 types of frozen soil can also be approximately classified into different types. For instance, clay soil has small particles and strong water-binding ability. Hence, it belongs to high water-holding capacity soil (HWHCS). The gravel soil and sand, by contrast, have more coarse particles and readily lose moisture. Therefore, it can be distinguished as low water-holding capacity soil (LWHCS). The water-holding capacity of silt and fully weathered rock is mediocre, which can be marked as medium water holding capacity soil (MWHCS). Therefore, to obtain the variation features and prediction models of soil thermal conductivity, the typical specimens in the QTEC are classified as three clusters according to their thermal conductivity probability distribution and water-holding capacity (as shown in Table 2).

**Table 2.** Classification of testing specimens.

| Cluster | Soil Types |
|---------|------------|
| HWHCS | clay, silty clay |
| MWHCS | silt, silty sand, weathered mudstone, weathered phyllite, weathered slate |
| LWHCS | medium fine sand, gravel sand, boulder, breccia, gravel |

*4.2. Thermal Conductivity Probability Distribution Characteristics for Three Clusters*

The average value and probability distribution of thermal conductivity of three soil clusters is shown in Table 3 and Figure 7. It can be seen that the roughly thermal conductivity magnitude order is LWHCS, MWHCS and HWHCS. The average thermal conductivity of unfrozen clusters ($\overline{\lambda}_u$) is 1.822, 1.556 and 1.418 W/(m·K). Among the main probability range (20%~80%), the thermal conductivity value of each unfrozen cluster varies between 1.518~2.171, 1.311~1.816, 1.193~1.605 W/(m·K). Whereas the corresponding average values are 2.258, 1.999 and 1.822 W/(m·K), and the high-probability range is 1.811~2.709, 1.611~2.334 and 1.493~2.154 W/(m·K) for the frozen clusters ($\overline{\lambda}_f$), respectively (see Table S2 in Supplementary Materials). This is because thermal conductivity of soil's mineral skeleton and the particle size of LWHCS are relatively larger than those of MWHCS and HWHCS. Meanwhile, based on the average dry density and water content data in Table 3, it can be inferred theoretically that the average density ($\rho$, $\rho = \rho_d(1 + w)$) of LWHCS is still large one under natural condition. So the test results show that the average values and main probability range of thermal conductivity of LWHCS is generally larger than those of MWHCS and HWHCS.

**Table 3.** Average thermal conductivity for three soil clusters.

| Cluster | $\bar{\rho}_d$(g/cm$^3$) | $\bar{w}$ (%) | $\bar{\lambda}_u$/W·(m$^{-1}$·K$^{-1}$) | $\bar{\lambda}_f$/W·(m$^{-1}$·K$^{-1}$) |
|---|---|---|---|---|
| HWHCS | 1.648 | 28.730 | 1.418 | 1.822 |
| MWHCS | 1.813 | 23.996 | 1.556 | 1.999 |
| LWHCS | 1.936 | 17.464 | 1.822 | 2.258 |

**Figure 7.** Probability distribution of thermal conductivity of three soil clusters.

*4.3. Distribution of Dry Density and Water Content, and Effect on Thermal Conductivity*

4.3.1. Dry Density and Water Content Distribution Characteristics

The dry density ($\rho_d$) and water content ($w$) of three soil clusters are statistically calculated, and the cumulative proportion is shown in Figure 8. It shows that the HWHCS has relatively lower dry density and higher water content among all three soil clusters. The main dry density distribution interval of the three clusters (probability within 20%~80%) is 1.34~1.78, 1.44~1.86, and 1.57~2.03 g/cm$^3$, and the mean values is 1.648, 1.813, and 1.936 g/cm$^3$ respectively. Likewise, the main water content distribution interval for every cluster is 15.32%~35.88%, 12.2%~30.6% and 7.7%~25.7%, and the average values are 28.73%, 24 % and 17.46%, respectively.

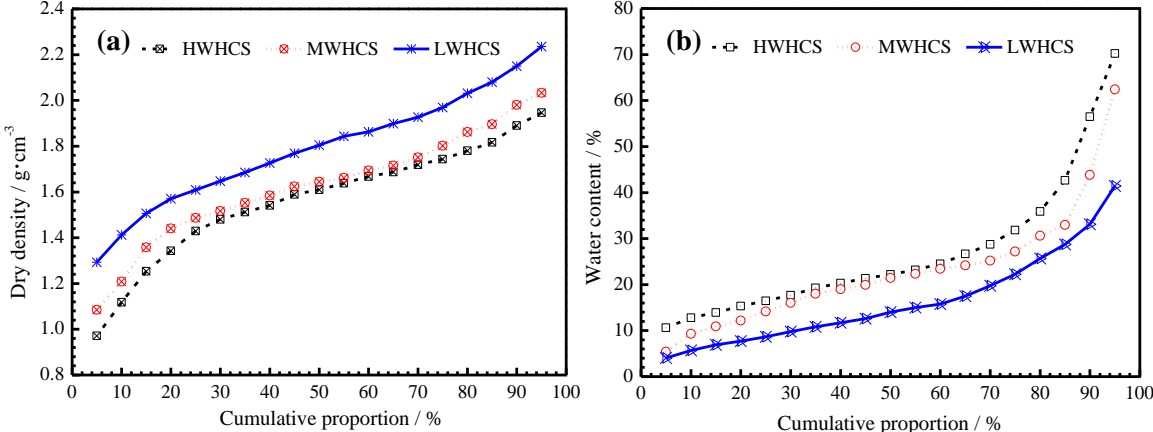

**Figure 8.** (**a**) Dry density and (**b**) water content cumulative proportion of three soil clusters.

4.3.2. Effect of Dry Density and Water Content on Thermal Conductivity

Preliminary research results show that both dry density and water content are significant influencing factors of soil's thermal conductivity [7,8]. Generally, it is considered that dry density and

water content have positive correlation with soil thermal conductivity. However, the test results of thermal conductivity of three soil clusters show that this conclusion is sometimes invalid. Thermal conductivity distribution of different dry densities and water content are shown in Figures 9 and 10. It shows that the thermal conductivities of three soil clusters discretely distributes at different dry density and water content intervals. In most cases, the thermal conductivity has no obvious correlations with dry density or water content. This is because the combination of soil dry density and water content parameters is diverse under natural environmental conditions, and as a complex multiphase mixture, soil thermal conductivity is influenced by many other factors. Meanwhile, it should be noted that the thermal conductivity shows an approximate increasing trend for certain cases, such as unfrozen LWHCS with increasing dry density, and frozen LWHCS and MWHCS with increasing water content.

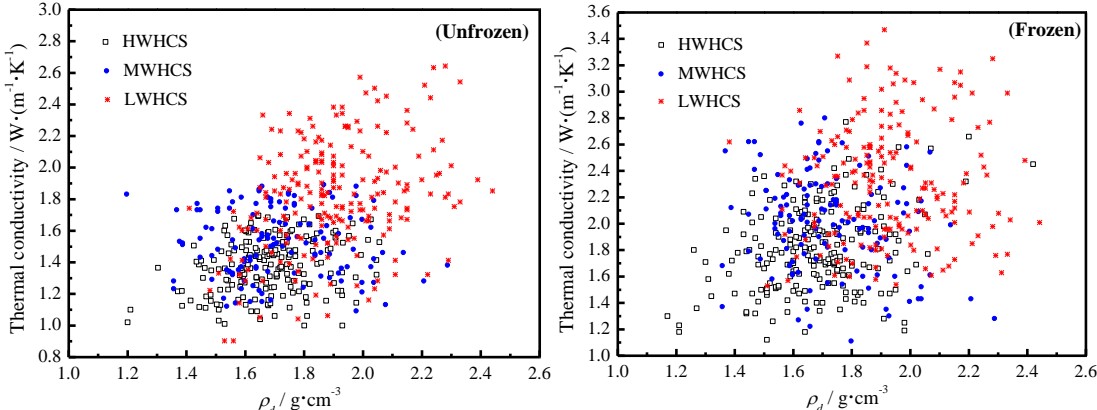

**Figure 9.** Thermal conductivities of three soil clusters with different dry density.

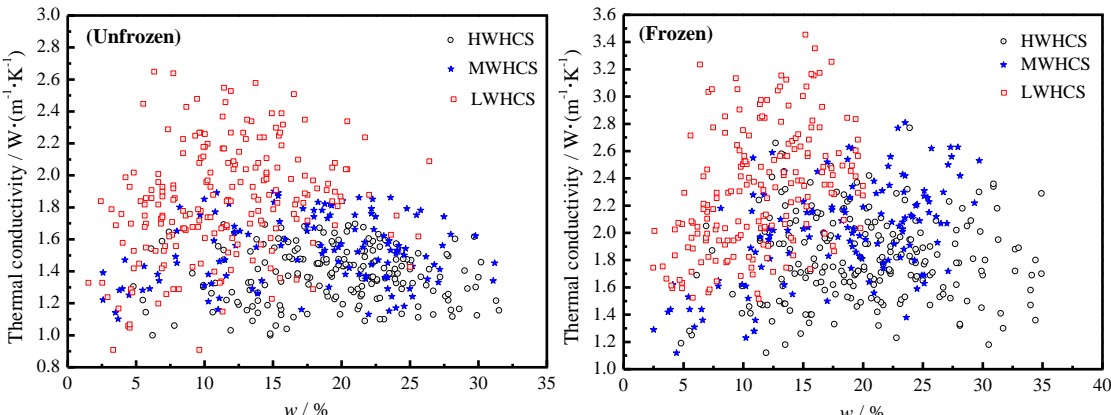

**Figure 10.** Thermal conductivities of three soil clusters with different water content.

## 5. Thermal Conductivity Prediction for Three Soil Clusters

### 5.1. Thermal Conductivity Prediction Models

#### 5.1.1. Binary Fitting

The previous research indicates that dry density and water content have obvious positive correlation with thermal conductivity. Therefore, the curve estimation of the fitting relationship between thermal conductivity, dry density and water content has been taken and exponential fitting formulas for three soil clusters are obtained:

$$\lambda = a_1 e^{b_1 \rho_d + c_1 w} \tag{2}$$

where $a_1$, $b_1$ and $c_1$ are the fitting coefficients of the equation.

It is also found that the influence of water content on thermal conductivity is inconsistent with different dry density. When the dry density is small, increasing of water cannot play a significant role in increasing soil connectivity as well as increasing dry density. On the contrary, the increasing of water is the most effective approach to promote soil thermal conductivity under the high dry density condition. For this reason, the fitting formulas are stepwise fitted according to their dry density. The detailed fitting parameters of three soil clusters are listed in the Table 4.

**Table 4.** Binary fitting parameters for three soil clusters.

| Soil Type | Operation Condition | Application Range | Parameters | | |
|---|---|---|---|---|---|
| | | | $a_1$ | $b_1$ | $c_1$ |
| HWHCS | Unfrozen | $\rho_d \leq 1.7 \text{ g/cm}^3$ | 0.3441 | 0.7152 | 0.6865 |
| | | $\rho_d > 1.7 \text{ g/cm}^3$ | 0.4158 | 0.5942 | 0.7932 |
| | Frozen | $\rho_d \leq 1.7 \text{ g/cm}^3$ | 0.3385 | 0.8262 | 1.1973 |
| | | $\rho_d > 1.7 \text{ g/cm}^3$ | 0.4569 | 0.6011 | 1.4366 |
| MWHCS | Unfrozen | $\rho_d \leq 1.7 \text{ g/cm}^3$ | 0.5338 | 0.5586 | 0.751 |
| | | $\rho_d > 1.7 \text{ g/cm}^3$ | 0.8478 | 0.2332 | 1.44 |
| | Frozen | $\rho_d \leq 1.7 \text{ g/cm}^3$ | 0.4906 | 0.586 | 1.9327 |
| | | $\rho_d > 1.7 \text{ g/cm}^3$ | 0.88 | 0.2748 | 1.7438 |
| LWHCS | Unfrozen | $\rho_d \leq 1.9 \text{ g/cm}^3$ | 0.298 | 0.8404 | 1.4741 |
| | | $\rho_d > 1.9 \text{ g/cm}^3$ | 0.6675 | 0.3675 | 2.912 |
| | Frozen | $\rho_d \leq 1.9 \text{ g/cm}^3$ | 0.4351 | 0.6979 | 2.366 |
| | | $\rho_d > 1.9 \text{ g/cm}^3$ | 0.9374 | 0.2549 | 3.8056 |

### 5.1.2. Radial Basis Function (RBF) Neural Network

The radial basis function (RBF) neural network has the characteristics of simple structure, fast learning speed and good convergence, and is widely used in the field of nonlinear function approximation [46,47]. Thus, this method has been applied for three clusters' thermal conductivity prediction. The dry density and water content is respectively used as the input layer of the RBF neural network, and the thermal conductivity is used as the output layer. The settings of the neural network structure in JMP Pro 13.0 software mainly include the activation function, the number of hidden layers, and the number of nodes in each layer. Considering the calculation accuracy and amount, the radial Gaussian function is selected as the activation function, the number of hidden layers is set as 2, and the number of nodes in each layer is 10. The established relationship diagram of the RBF neural network is shown in Figure 11.

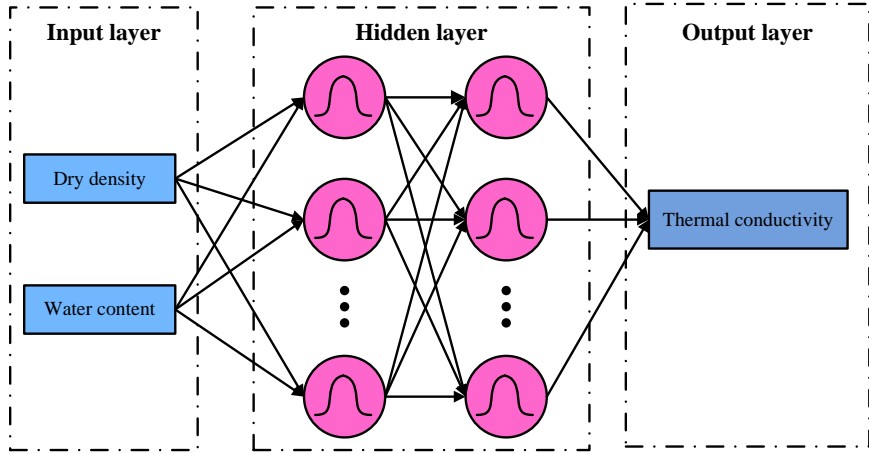

**Figure 11.** Schematic diagram of radial basis function (RBF) neural network.

Besides, all of thermal conductivity measurement results of three soil clusters are randomly divided into 9:1 ratio; 90% of the sample data were used for training samples of the neural network model, and the remaining 10% were used as the validation of RBF neural network predictive model. Additionally, based on the feedback of training results, small samples with larger deviations are eliminated in the calculation to improve the prediction accuracy. Figure 12 is the predictive values of unfrozen MWHCS. It can be seen that the predicted results of RBF neural network method are in good agreement with the experimental results ($R^2 = 0.82$).

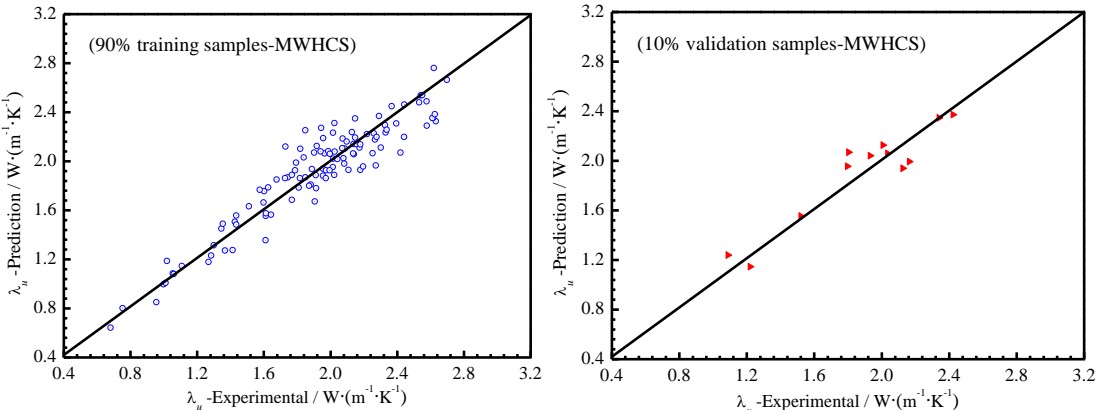

**Figure 12.** Predictive values of unfrozen MWHCS using RBF neural network method.

### 5.1.3. Ternary Fitting for Frozen Soil

The soil distribution in the QTEC is diverse and can be divided into different types according to geotechnical specifications. However, the information of mineral composition and particle size distribution cannot be fully specified by the geotechnical engineering classification method. The dry density and water content are not the exclusive influence factors for soil thermal conductivity. Hence, this will lead to certain deviation for predicting the thermal conductivity of frozen soil only using the above two indicators.

Furthermore, compared to the frozen soil sample, the unfrozen soil sample is relatively easier to prepare and test, and its thermal conductivity measurement result naturally implies the information of specimen's mineral composition and particle size distribution. The correlation analysis proves strong positive correlations between frozen and unfrozen thermal conductivity either. So the ternary fitting formula, using dry density, water content and an unfrozen specimen's thermal conductivity as fitting parameters, of frozen sample's thermal conductivity has been developed in this work:

$$\lambda_f = a_2 \lambda_u + b_2 + c_2 \ln(w) + d_2 \ln(\rho_d) \tag{3}$$

where $a_2$, $b_2$, $c_2$ and $d_2$ are the fitting coefficients of the formula (shown in Table 5).

**Table 5.** Ternary fitting parameters for three soil clusters.

| Parameters | HWHCS | MWHCS | LWHCS |
|:---:|:---:|:---:|:---:|
| $a_2$ | 1.254 | 1.167 | 1.178 |
| $b_2$ | 0.413 | 0.599 | 0.634 |
| $c_2$ | 0.273 | 0.332 | 0.312 |
| $d_2$ | 0.2 | 0.354 | 0.22 |

### 5.2. Comparison of Thermal Conductivity Prediction Models

The predictive results of three prediction models are shown in Figure 13. It should be noted that the prediction models of different soil clusters remove some abnormal sample points for feasible

calculation, which account for 2%~15% of the total samples number (seen in Table 5). Moreover, for the large quantity of sample points and clear exhibition, only 1/3 randomly selected sample points are shown in Figure 13. It can be seen that most sample points are within the ±10% error bars for all three prediction models, which proves their predictive availability and engineering application values. Furthermore, the sample points of the binary fitting method are mainly concentrated in the medium thermal conductivity range, and there is almost no distribution at both high- and low-value intervals. Conversely, the sample distribution range of the RBF neural network method and the ternary fitting method is much broader, which is distributed throughout the thermal conductivity interval. Compared with the quantitative proportion of predictive values deviating severely from experimental values, it can be found that the binary fitting method is much more than that of the RBF neural network method and the ternary fitting method.

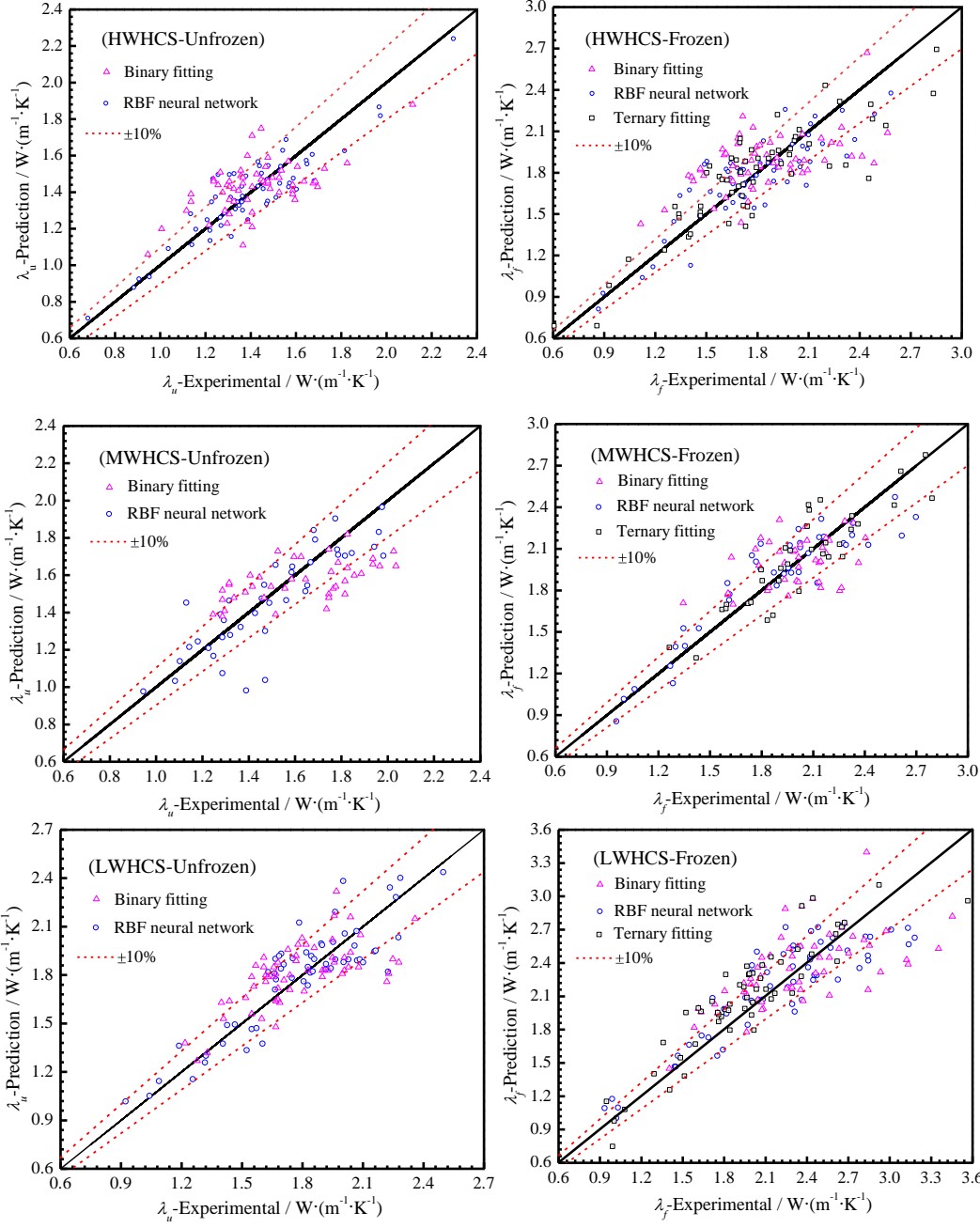

**Figure 13.** Predictive results of three prediction models.

The $R^2$ and quantitative proportion of relative error within $\pm10\%$ ($P_{\pm10\%}$) for three prediction models is shown in Tables 6 and 7. It can be seen that the fitting degree of the RBF neural network method and the ternary fitting method is much higher than that of the binary fitting method. The maximum $R^2$ and $P_{\pm10\%}$ of the ternary fitting method reaches up to 0.88 and 74.5%, while binary fitting method is only 0.561 and 39.6%. Besides, the abnormal sample rejection ratio of three predictive models is 10%~15%, 10% and 2%~3%, respectively. This demonstrates that the ternary fitting is the most accurate thermal conductivity prediction method. However, the ternary fitting method has some methodological limitations: (1) the fitting formulas are based on a large amount of test data, which cannot completely avoid high costs and a lengthy thermal conductivity experiment; (2) it can only be used to predict thermal conductivity of frozen soils, which limits its application scope. The RBF neural network method, meanwhile, has comparative $R^2$ and $P_{\pm10\%}$ as well as the ternary fitting method, and it can also be used for unfrozen soils. It also can be noted that the $R^2$ and $P_{\pm10\%}$ of MWHCS is usually lower than that of HWHCS and LWHCS with binary or ternary fitting models, while the prediction results of the RBF neural network method for MWHCS is more accurate. Therefore, it can be concluded that the RBF neural network method can better identify soil's complex composition characteristics and be the optimal thermal conductivity prediction method.

**Table 6.** $R^2$ for three prediction models.

| Prediction Model | Abnormal Sample Rejection Ratio | $R^2$ | | | | | |
| --- | --- | --- | --- | --- | --- | --- | --- |
| | | HWHCS | | MWHCS | | LWHCS | |
| | | Unfrozen | Frozen | Unfrozen | Frozen | Unfrozen | Frozen |
| Binary fitting | 10%~15% | 0.561 | 0.512 | 0.475 | 0.432 | 0.532 | 0.487 |
| RBF neural network | 10% | 0.79 | 0.79 | 0.82 | 0.75 | 0.81 | 0.8 |
| Ternary fitting | 2%~3% | — | 0.77 | — | 0.8 | — | 0.88 |

**Table 7.** Quantitative proportion of relative error within $\pm10\%$ for three prediction models.

| Prediction Model | Fitting Parameters | $P_{\pm10\%}$ (%) | | | | | |
| --- | --- | --- | --- | --- | --- | --- | --- |
| | | HWHCS | | MWHCS | | LWHCS | |
| | | Unfrozen | Frozen | Unfrozen | Frozen | Unfrozen | Frozen |
| Binary fitting | $w, \rho_d$ | 59.8 | 43.6 | 42.5 | 39.6 | 60.3 | 54.6 |
| RBF neural network | $w, \rho_d$ | 77.9 | 65.1 | 81.4 | 70.1 | 71.6 | 63.2 |
| Ternary fitting | $w, \rho_d, \lambda_u$ | — | 65.7 | — | 70.5 | — | 74.5 |

## 6. Discussion

### 6.1. Error Analysis of Thermal Conductivity Prediction Models

To compare the precisions of three prediction models, the mean absolute percent error (MAPE) of different thermal conductivity intervals has been calculated and analyzed:

$$MAPE = \sum_{i=1}^{n} \left| \frac{\lambda_{ei} - \lambda_{pi}}{\lambda_{ei}} \right| \times \frac{100}{n} \tag{4}$$

where $\lambda_{ei}$ and $\lambda_{pi}$ is the experimental and predictive thermal conductivity respectively; $n$ is the total sample numbers of certain thermal conductivity value.

The MAPE distribution of HWHCS is shown in Figure 14. This shows that the MAPE distribution curve of the binary fitting method is steeper. Its bilateral MAPE can be up to 21% and 16% for frozen and unfrozen soils, and its small MAPE interval is narrow which restricts its application in the large or small thermal conductivity prediction. While the MAPE of the RBF neural network method

and ternary fitting method are much lower, comparatively, and can be applied to the entire thermal conductivity distribution interval. Additionally, it should be mentioned that the relative accurate thermal conductivity predictive intervals (MAPE < 10%) of binary fitting method for three clusters is 1.24~1.62 W/(m · K), 1.4~1.7 W/(m · K), 1.54~2.06 W/(m · K) (unfrozen) and 1.7~2.08 W/(m · K), 1.9~2.3 W/(m · K), 1.96~2.64 W/(m · K) (frozen), respectively, which is roughly overlapped with main thermal conductivity probability range as shown in Figure 7. Therefore, considering easily accessed dry density and water content data, the binary fitting method definitely has certain application value in engineering.

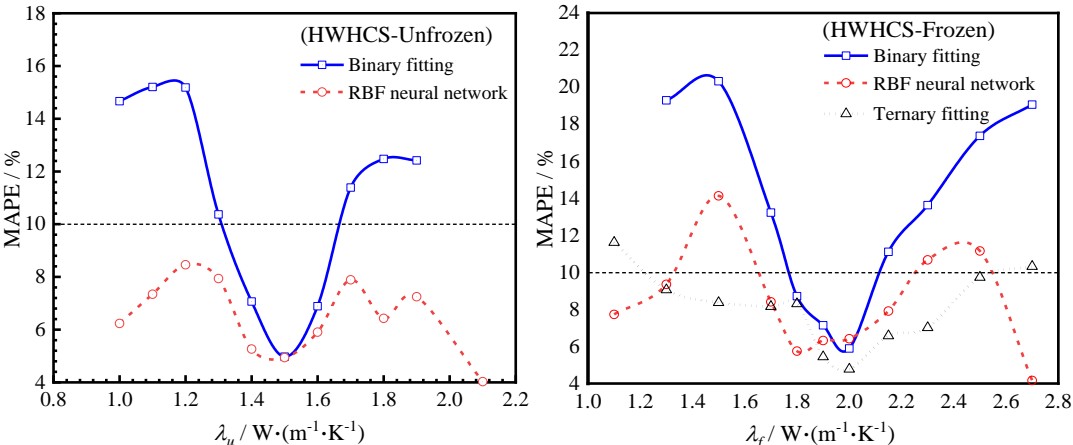

**Figure 14.** Mean absolute percent error (MAPE) distribution of HWHCS.

*6.2. Prediction Effect and Influence Factors of RBF Neural Network Model*

As everyone knows, soil is a complex multiphase mixture and its thermal conductivity is influenced by many factors [7,8]. Among various impact parameters, dry density and water content are the most important and analyzed factors. A generally accepted concept is that soil thermal conductivity increases as the dry density or water content increase. However, as shown in Figure 15, dry density as well as water content do not have strong positive correlations with soil thermal conductivity under specific natural conditions. The thermal conductivity of HWHCS distributes randomly and sparsely under different dry density and water content, and it seems hard to find a suitable function describing this phenomenon. Thus, the binary fitting formula can only take the form of a piecewise function to promote prediction accuracy. However, the $R^2$ of fitting formulas are still very low (0.561 for unfrozen HWHCS). Herein, it can be concluded that the formula fitting method based on incomplete parameters may not be effective for soil thermal conductivity prediction, especially with multiple soil types and complex compositions.

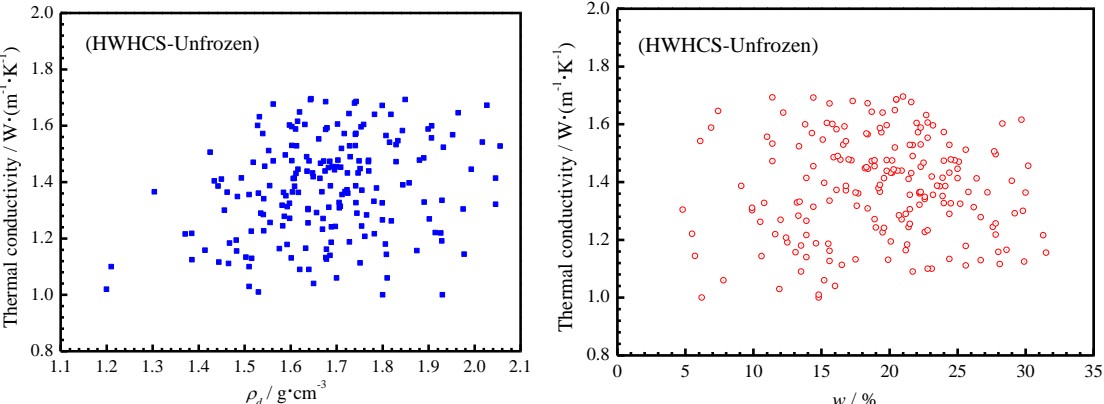

**Figure 15.** Thermal conductivities of HWHCS with different dry density and water content.

The RBF neural network method is capable of capturing and depicting features and sub-features embedded in a large amount of data, and of obtaining a definite output [39]. The output of RBF neural network is produced by mapping distances between input vectors and center vectors (hidden layer) to output through a non-linear kernel. The most attractive characteristic of a RBF neural network algorithm is that it can capture non-linear and complex relationship among parameters and build a feasible output prediction model using imperfect or incomplete data. Considering for these advantages, the RBF neural network method has been adopted for the prediction of soil thermal conductivity using dry density and water content as input parameters. It can be seen in Figure 13 and Table 6 that the RBF neural network method has achieved good prediction results. The $R^2$ of unfrozen HWHCS, MWHCS, and LWHCS is 0.79, 0.82 and 0.81, which is much better than the binary fitting results.

Moreover, in addition to developing and optimizing the algorithm itself, the neural network methods need a large amount of experimental data to obtain reasonable prediction results. Fortunately, nearly 1000 test data successively avoid the overfitting problem during training computation of a neural network. Therefore, despite input parameters being only part of the influencing factors, advanced algorithms and sufficient experimental data greatly improve the prediction accuracy of the RBF neural network model.

## 7. Conclusions

In the present work, the thermal conductivity of 638 unfrozen specimens and 860 frozen specimens sourced from the Qinghai–Tibet expressway geological exploration project has been measured by the TPS method. Then, based on the cluster analysis of typical soils, the probability distribution of dry density, water content and thermal conductivity, and the influence of the dry density and water content on the thermal conductivity of three soil clusters, have been investigated. Finally, using water content, dry density and unfrozen soil thermal conductivity as fitting parameters, binary fitting, RBF neural network and ternary fitting calculation models have been developed and compared for the three soil clusters' thermal conductivity prediction. The results show that:

(1) The particle size and intrinsic heat-conducting capacity of the soil skeleton have a significant influence on the soil thermal conductivity, and the typical specimens in the QTEC can be classified as three clusters according to their thermal conductivity probability distribution and water-holding capacity.

(2) Dry density as well as water content sometimes have statistically low correlation with thermal conductivity of natural soil samples, especially for multiple soil types and complex compositions.

(3) Both the RBF neural network method and the ternary fitting method have favorable prediction accuracy and wide application range, which can be used to predict the soil thermal conductivity in the QTEC. As the ternary fitting method can only be used for frozen soils, the RBF neural network method is considered as the optimal prediction approach.

(4) The prediction effect and error analysis of the prediction results indicate that the accuracy of the binary fitting method is relatively poor compared with the RBF neural network and the ternary fitting method, but the main body interval of the thermal conductivity distribution can still meet general engineering requirements.

**Supplementary Materials:** The following are available online at http://www.mdpi.com/2076-3417/10/7/2476/s1, Table S1: Experimental database of 12 soil types; Table S2: Experimental database of 3 soil clusters.

**Author Contributions:** Conceptualization, Z.-Y.L. and F.-Q.C.; Writing—original Draft: Z.-Y.L. and F.-Q.C.; Writin—Review and Editing: Z.-Y.L. and J.-B.C.; Investigation: F.-Q.C., J.-B.C. and Y.-H.D.; Resources: Z.-Y.L. and L.J.; Data Curation: F.-Q.C., H.P.; Formal analysis: Y.-H.D. and L.J.; Validation: Y.-H.D.; All authors have read and agree to the published version of the manuscript.

**Funding:** This research was supported by the National Science Foundation of China (Grant No. 41502292, 51574037), the Applied Fundamental Research Project of China Communications Construction Co., Ltd. (No. 2018-ZJKJ-PTJS03), and the Project funded by China Postdoctoral Science Foundation (NO. 2014M560739).

**Conflicts of Interest:** The authors declare no conflict of interest.

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
