# Peer review of "Experimental Test and Prediction Model of Soil Thermal Conductivity in Permafrost Regions"

_applsci, doi:10.3390/app10072476_

Round 1
Reviewer 1 Report
Please find the attachment for review results.
Although the authors performed extensive experiments and analysis, the manuscript lacks of physical explanation of their results. Therefore, the reviewer strongly recommend the authors to add additional discussion section before conclusion for the physical explanation between correlations among thermal conductivity, water content, and dry density for different soil types and freezing/nonfreezing conditions.

Author Response
Comment 1: The authors performed experiments to find correlation between dry density, thermal conductivity, and water contents for different types of soils under frozen and unfrozen soils. In addition, neural network was implemented to predict thermal conductivity using other parameters.
The correlations among three parameters of dry density, thermal conductivity, and water contents are very low. However, the authors tried to improve prediction of thermal conductivities using neural network. In reality, physical explanation of correlations of dry density, thermal conductivity, and water contents is very important. If the results were much improved in prediction of thermal conductivity using dry density and water contents, the authors should physically explain the reason of its improvement. Therefore, the reviewer suggests the author to add Discussion section for physical reasons of the effects of dry density and water content on thermal conductivity for both frozen and unfrozen soils based on the results of neural network. The discussion section and be placed in front of conclusions.
Response: Thanks for the reviewers’ suggestions. The discussion section has been added in the revised manuscript. We analyzed the relations among soil thermal conductivity, dry density and water content, discussed the prediction effect of binary fitting method, and finally explained the physical reasons of prediction accuracy improvement of RBF neural network model.
Comment 2: The length of abstract is long and contents contains redundant information.
Response: The abstract have been thoroughly revised, some redundant sentences have been deleted accordingly.
Comment 3: line 28-29: revise the sentence to improve clarity.
Response: We add the explanation of R2 and (please check pdf) in revised paper. The sentence has been revised as “The maximum determine coefficient (R2) and quantitative proportion of relative error within ±10% () of each prediction model reaches up to 0.82, 0.88, 81.4% and 74.5% respectively”
Comment 4: In the introduction, do not reuse the sentences in abstract.
Response: The abstract and introduction have been thoroughly revised, and the identical sentences have been deleted or revised.
Comment 5: line 59: "Zhang et al [19]" should be "Zhang et al. [19]"
Response: They have been revised accordingly.
Comment 6: line 71: "kr" should be explained briefly.
Response: “kr” has been revised as “Ke”, a briefly explanation “Ke, which was the function of thermal conductivity of soils, dry soils and saturated soils” is added in the revised manuscript.
Comment 7: line 77-78: numerate "more soil parameters of ....."
Response: we add a short explanation “such as soli texture-dependent parameter, quartz content, fluid network connectivity, etc.” after above sentence.
Comment 8: line 84: Indicate nation at which Golmud and Lhasa are located. And other locations...
Response: The sentence has been revised as “The QTEC is defined as the engineering channel from Golmud to Lhasa in the western China”
Comment 9: line 88: write in what aspect "complex either" means
Response: The sentence has been revised as “and with changeable terrain structure, frequent geological disasters, cold and hypoxic climate, its engineering geological conditions are extremely harsh either.”
Comment 10: "3.1.1 Source of test samples" Add details of sampling methods and sample preparation.
Response: The following sentences “The drilling operation is implemented by truck-mounted geological exploration rig, and the diameter of sampler is 108 mm. After taking out from sampler, the soil sample is immediately loaded into a 0.6m height marked sample vessel to avoid external disturbance.” have been added in the revised manuscript to specify the field sampling method and sample preparation.
Comment 11: Figure 2: "silty caly" should be "silty clay"
Response: They have been revised accordingly.
Comment 12: "3.2 Experimental methods" Add principal of thermal conductivity measurement using the Thermal Conductivity Analyzer
Response: The transient plane source (TPS) method is applied in the thermal conductivity measurement. The basic principal of TPS method is specified in the revised manuscript.
Comment 13: line 153: Provide details of field measurements of dry density and water content
Response: The dry density and water content is tested by the suspending weigh method and oven drying method respectively. And it has been added in the revised manuscript.
Comment 14: line 163-164: It is worried that the test data are excluded for more than 5% error. At least the authors should indicate the uncertainties of raw data and reasons of the high error. "For every specimen, the thermal conductivity measurement conducts twice or more times if the measuring error of testing results before and after is larger than 5%."
Response: The Hot Disk 1500s needs to define appropriate test time and power parameters to get reasonable test results. While specimen with different soil type, dry density or water content, the above 2 parameters are different either. Under the proper parameter settings, the measuring error of testing results is usually very small (<5%), on the contrary, the measuring error will be quite large as the parameter settings is improper. For every specimen, the thermal conductivity measurement conducts twice and takes the average of the two test results as the experimental result. But if the measuring error is larger than 5% due to improper parameter settings, we conduct multiple experiments to obtain reasonable test results. The corresponding sentences have been revised in the manuscript.
Comment 15: At the bottom of line 184, write the reason of differences of thermal conductivities with respect to soil types. For example, write reason of "why the clay samples have smaller thermal conductivity?"
Response: The explanation on the differences of thermal conductivities with respect to soil types has been added in the revised manuscript.
Comment 16: Figure 8: write the full and correct titles of axes
Response: They have been revised accordingly.
Comment 17: Figures 9 and 10: correlation coefficient or R^2 values should be provided. There seems no correlation between two parameters in the figures. The reviewer can not find points of drawing arrow lines when it is less correlated.
Response: Thanks for the reviewers’ suggestions. According to reviewer’s suggestion, Figures 9 and 10 has been reprocessed. The thermal conductivity of three soil clusters (frozen and unfrozen) with different dry density and water content has been given in the revised manuscript.
Comment 18: Provide plots showing the comparison between measured thermal conductivity and predicted thermal conductivity using equation (2).
Response: The comparisons between the measured thermal conductivity and the binary fitting method predicted value of all three clusters have been given in Fig. 13
We appreciate for the Reviewer’s warm work earnestly, and hope that the correction will meet with approval. Once again, thank you very much for your comments and suggestions.
With personal regards,
Sincerely yours,
Zhiyun Liu

Reviewer 2 Report
see the attachment

Author Response
Comment 1: Authors used 860 soil samples collected alone QTEC and applied three different fitting methods to predict thermal conductivity of the different soil types. These methods showed some interesting probability distributions. The counter intuitive part was that low soil moisture sample yield higher thermal conductivities. I would expect the opposite (see my comment on Figure 7b). It was not clear to me, if authors froze dry samples, then why did thermal conductivity change? The frozen conductivity of a soil sample changes only when water is present. Was water added to the samples before freezing or during freezing? This is an important step and requires clarification. In addition, Figure 9 and 10 do not show any correlation and I would not spend time trying to convince a reader that the correlations exist. I included my recommendation on performing some cluster analysis that can help to understand the clustering of the thermal conductivities (see Fig 9 and 10 comment). The manuscript needs to be reviewed by the proficient English language reader before sending it publication.
Response: The responses to the reviewers' questions are as follows:1) The soil thermal conductivity is mainly determined by the intrinsic heat-conducting capacities of solid and liquid phases, and the average heat transfer cross section area. The corresponding soil types of HWHCS are clay and silty clay, while the corresponding soil types of LWHCS are medium fine sand, gravel sand, boulder, breccia, gravel. The thermal conductivity of soil‘s mineral skeleton and particle size of LWHCS are both larger than HWHCS. Meanwhile, it can be inferred theoretically from Table 3 that the average density (ρ=ρd*(1+w)) of LWHCS is also larger than HWHCS under natural condition. So the test results show that the average values and main probability range of thermal conductivity of HWHCS is lower than that of LWHCS. Based on above explanation, the sentences of L216~218 have also been revised. 2) As shown in Fig. 3, the specimen is always placed in the polymethyl methacrylate sample ring and covered with plastic film during the measuring process. When the measurement of unfrozen soil thermal conductivity is finished, the specimens are placed in a freezing cabinet immediately and freeze for 24 hours. After the freezing process is complete, the measurement of frozen soil thermal conductivity has been taken successively. So the drying and water adding experimental procedures have not been taken before or during freezing process. 3) According to reviewer’s suggestion, Fig. 9 and 10 have been reprocessed. The thermal conductivity distribution of three soil clusters with different dry density and water content has been given in the revised manuscript. The corresponding explanations of the Fig.9 and 10 are also revised. 4) We revised the whole manuscript carefully to avoid language errors. In addition, we asked several colleagues who are skilled authors to check the English. Thanks for the reviewers’ valuable comments and these suggestions are very helpful for improving our manuscript.
Comment 2: L25. “But this doesn’t …” rephrase that sentence. State explicitly what do you mean by‘this’? In the end of the Abstract add a sentence on the application/impact of this study.
Response: the sentence has been revised as “dry density as well as water content sometimes doesn’t have strong positive correlation with thermal conductivity of natural soil samples, especially for multiple soil types and complex compositions.” And the sentence-“This study can contribute to the construction and maintenance of engineering applications in permafrost regions.” has been added at the end of abstract to specify the research implication.
Comment 3: L63 in addition to the listed previous studies, there were several attempts to recover soil thermal conductivities using inverse modeling (e.g. Nicolsky et al., 2009, Jafarov et al., 2020).
Response: The reference works on the inverse modeling has been added in the introduction part (Line 76-78) and reference list.
Comment 4: L116 Rephrase the sentence.
Response: The sentence has been revised accordingly.
Comment 5: L128 The sentence does not make sense. Rephrase.
Response: The sentence has been revised as “To guarantee the experiment progress and data representativeness, the specimens with large dry density, high water content, and number of samples less than 15 have not been measured in the test.”
Comment 6: L130 Instead of calling categories I suggest to call them soil types. Then it will be clearer and make more sense to reader.
Response: The “categories” has been revised as “soil types” or “types” in the revised manuscript.
Comment 7: Table 1. It would be nice to plot the distribution of the of soil types and add 7 subplots to the figure 2 showing distribution of the soil type for each depth layer.
Response: The Table 1 and Fig.2 have been revised as reviewer’s suggestions. The detailed soil types and numbers of 7 depth ranges for every soil type have been given in the revised manuscript.
Comment 8: L138 Not sequential referencing. Starting from Fig3c not describing Fig 3a and 3b. Put change c and a and make sure that you describe each plot in sequential order. Also, you did not mention Fig3a.
Response: Fig. 3 has been reordered. The corresponding descriptions for each plot are revised in sequential order. And the description sentence for “Soil sample library” has also been added in the revised manuscript.
Comment 9: L163 From the description it is not clear if there was a water added during freezing. Not
clear why each sample conducts 2 or more times higher after freezing it was dry prior to freeing?
Response: 1) As shown in Fig. 3, the specimen is always placed in the polymethyl methacrylate sample ring and covered with plastic film during the measuring process. When the measurement of unfrozen soil thermal conductivity is finished, the specimens are placed in a freezing cabinet immediately and freeze for 24 hours. After the freezing process is complete, the measurement of frozen soil thermal conductivity has been taken successively. So the drying and water adding experimental procedures have not been taken before or during freezing process. 2) As we mentioned in the manuscript, although the Hot Disk 1500s has good measurement accuracy, it still needs to define appropriate test time and power parameters to get reasonable test results. While specimen with different soil type, dry density or water content, the above 2 parameters are different either. For every specimen, the thermal conductivity measurement conducts twice and takes the average of the two test results as the experimental result. But if the measuring error is larger than 5% due to improper parameter settings, we conduct multiple experiments to obtain reasonable test results. The corresponding sentences have been revised in the manuscript.
Comment 10: L176 What is cohesive soil? You need to define it first.
Response: The “cohesive soil” has been revised as “clay and silty clay soil” in the revised manuscript.
Comment 11: L179 Specify mudstone soils
Response: The “mudstone” has been revised as “weathered mudstone” in the revised manuscript.
Comment 12: Figure 5a has 5 and 5b has 6 soil type. You are missing one. Caption. Change ‘categories’ to soil types.
Response: Fig. 5a has been revised accordingly. The thermal conductivity distribution of all 12 types unfrozen soil had been given in the manuscript. The “categories” in the figure caption has also been revised as “soil types”.
Comment 13: Similarly Figure 6 does not have 12 soil type missing 2.
Response: Fig. 6 has been revised as review’s suggestions. The thermal conductivity distribution of all 12 types frozen soil had been given in the revised manuscript.
Comment 14: Figure7b not sure why frozen thermal conductivity for HWHCS is lower that LWHCS. I would expect it to be opposite. Need to better explain that. L216-218 needs better explanation.
Response: The soil thermal conductivity is mainly determined by the intrinsic heat-conducting capacities of solid and liquid phases, and the average heat transfer cross section area. The corresponding soil types of HWHCS are clay and silty clay, while the corresponding soil types of LWHCS are medium fine sand, gravel sand, boulder, breccia, gravel. The thermal conductivity of soil‘s mineral skeleton and particle size of LWHCS are both larger than HWHCS. Meanwhile, it can be inferred theoretically from Table 3 that the average density (ρ=ρd*(1+w)) of LWHCS is also larger than HWHCS under natural condition. So the test results show that the average values and main probability range of thermal conductivity of HWHCS is lower than that of LWHCS. Based on above explanation, the sentences of L216~218 have also been revised.
Comment 15: L225 what is the meaning of the accumulated probability distribution?
Response: It has been revised as “cumulative proportion”, which is the proportion of samples with a dry density or water content in the range of 0~y to the total number of samples.
Comment 16: Figure 9 and 10 fitting straight line does not make much sense. You can fit and polynomial to that distribution. It is clear that there is no definite correlation. So, instead, I would suggest to perform some sort of clustering analysis and put all unfrozen (L,M,H) in one figure and all frozen in another. Use three different marks to differentiate them. Then that figure might give a better idea on clustering of different water content (L,M,H) soil. Start x and y axis from (0, 0).
Response: Thanks for the reviewers’ suggestions. According to reviewer’s suggestion, Fig. 9 and 10 have been reprocessed. The thermal conductivity distribution of three soil clusters with different dry density and water content has been given in the revised manuscript. The corresponding explanations of the Fig.9 and 10 are revised too.
Comment 17: Table 4. Where the 1.7 and 1.9 numbers come from? How are changes in those numbers would affect the RBF results?
Response: In the research, we tried to develop empirical fitting formula of thermal conductivity using dry density and water content as fitting parameters. However, just as shown in Fig.9 and 10, the thermal conductivity distributed dispersed under different dry density or water content. For better fitting results, it is found that the dry density-dependent piecewise fitting formula can obtain highest R2 after many attempts. The segmentation point 1.7 and 1.9 in Table 4 are the results of multiple trials. It is only used in the binary fitting and has nothing to do with RBF neural network results.
Comment 18: L296. ‘…roughly…’? Rephrase that sentence.
Response: The sentence has been revised as “However, the information of mineral composition and particle size distribution can’t be fully specified by geotechnical engineering classification method.”
Comment 19:Table 5. where are a2,b2,c2,d2 come from? How are changes in those numbers would affect the results of the ternary fitting?
Response: The a2, b2, c2, d2, are the fitting coefficients of ternary fitting formula, which is computed by the SPSS software. It is obviously that the prediction accuracy of ternary fitting depends on the precision of above number.
Comment 20: Figure 13 add to the caption what are dashed red lines represent. Why do those lines widen towards the top?
Response: The dashed red lines represent the ±10% deviation values to the measured thermal conductivity. They are the absolute error lines, the difference will increase as the experimental value increase. So the lines widen towards the top. Fig. 13 has been revised as reviewer’s suggestion in the manuscript.
Comment 21:L335 What are the methodological limitation? Need to explain better.
Response: The methodological limitation means: On the one hand, the ternary fitting method need to get a lot of dry density, water content and unfrozen specimen’s thermal conductivity test data in advance, which can’t completely avoid high costs and long test time of thermal conductivity measurement; On the other hand, the ternary fitting method can only be used to predict thermal conductivity of frozen soils, which limits its application scope. Corresponding explanations are also given in the revised manuscript.
Comment 22: From figure 14 it looks like ternary fitting performed better that other two.
Response: As the reviewer pointed out, the ternary fitting is the most accurate thermal conductivity prediction method among three models. It can be seen in Fig. 14 that the MAPE of ternary fitting method is almost the lowest at the entire thermal conductivity distribution range. However, as we mentioned before, the ternary fitting method can only be used for frozen soils on the premise of unfrozen soils’ thermal conductivity acquisition, which means relatively high costs and narrow engineering application scope. While the RBF neural network method can used for both frozen and unfrozen soils, its prediction accuracy is very close to that of ternary fitting method. Furthermore, RBF neural network method only requires dry density and water content data as input parameters. The cost and difficulty of above two parameters are relative low in engineering practice. To sum up, it can be concluded that the RBF neural network method can be the optimal thermal conductivity prediction method.
Recommended references
Nicolsky, D.J., Romanovsky, V.E., Panteleev, G.G., 2009. Estimation of soil thermal properties using in-situ temperature measurements in the active layer and permafrost. Cold Reg. Sci. Technol. 55 (1), 120–129.
Jafarov, E. E., Harp, D. R., Coon, E. T., Dafflon, B., Tran, A. P., Atchley, A. L., Lin, Y., and Wilson, C. J.: Estimation of subsurface porosities and thermal conductivities of polygonal tundra by coupled inversion of electrical resistivity, temperature, and moisture content data, The Cryosphere, 14, 77–91, https://doi.org/10.5194/tc-14-77-2020, 2020.